# The Role of Immersive Virtual Reality Interventions in Pediatric Cerebral Palsy: A Systematic Review across Motor and Cognitive Domains

**DOI:** 10.3390/brainsci14050490

**Published:** 2024-05-12

**Authors:** Maria Grazia Maggio, Maria Chiara Valeri, Rosaria De Luca, Fulvia Di Iulio, Irene Ciancarelli, Morena De Francesco, Rocco Salvatore Calabrò, Giovanni Morone

**Affiliations:** 1IRCCS Centro Neurolesi “Bonino Pulejo”, S.S. 113, Contrada Casazza, 98124 Messina, Italy; mariagrazia.maggio@irccsme.it (M.G.M.); rosaria.deluca@irccsme.it (R.D.L.); morenadefrancesco00@gmail.com (M.D.F.); 2Department of Life, Health and Environmental Sciences, University of L’Aquila, 67100 L’Aquila, Italy; mariachiara.valeri@student.univaq.it (M.C.V.); irene.ciancarelli@univaq.it (I.C.); giovanni.morone@univaq.it (G.M.); 3IRCCS Santa Lucia Foundation, 00100 Rome, Italy; f.diiulio@hsantalucia.it; 4San Raffaele Institute of Sulmona, 67039 Sulmona, Italy

**Keywords:** cerebral palsy, cognitive outcomes, motor outcomes, virtual reality

## Abstract

**Background:** In recent years, new technologies have been applied in cerebral palsy. Among these, immersive virtual reality is one with promising motor and cognitive effects along with the reduced costs of its application. The level of immersion of the subject in the illusional world gives the feeling of being a real part of the virtual environment. This study aims to investigate the safety and the efficacy of immersive virtual reality in children affected by cerebral palsy. **Methods:** PubMed, Embase, Cochrane Database of Systematic Reviews, RehabData, and Web of Science were screened up to February 2023 to identify eligible clinical studies. **Results:** Out of 788, we included 15 studies involving CP patients. There was high heterogeneity in the outcomes considered, and the results showed non-inferiority to conventional therapy and initial additional benefits in comparison with conventional rehabilitation. **Conclusions:** Immersive virtual reality emerges as a pivotal technological tool in rehabilitation, seamlessly integrating with conventional therapy within CP rehabilitation programs. Indeed, it not only enhances motivation but significantly increases children’s functional capacity and abilities.

## 1. Introduction

Cerebral palsy (CP) consists of a group of permanent disorders in the development of movement and posture, causing limitations in activity. These disorders are attributed to non-progressive alterations in the fetal and infant brain. The motor impairments associated with CP are frequently accompanied by disturbances in sensation, perception, cognition, communication, and behavior, as well as epilepsy and secondary musculoskeletal problems [1]. According to recent research studies, there is a prevalence of CP in countries with advanced healthcare services, estimated at about 2–3 cases per thousand live births [2,3]. This rate seems to have remained relatively constant over the past decade. Certain populations of newborns are at particularly high risk, with an incidence over 70 times higher in infants weighing < 1500 g, mostly extremely premature infants. As of today, spastic forms account for approximately 80% of CP cases, dyskinetic forms for 10–15%, and ataxic forms for 5–10% [2]. The highest prevalence, 111.8/1000 live births, occurs in infants born extremely premature (<28 weeks of gestation), as reported by Neil Wimalasundera in 2016 [4]. Hemiplegia is present in 10% of severely preterm (born < 28 weeks) children with CP, 16% of preterm births (between 28 and 31 weeks), 34% of moderately preterm births (32–36 weeks), and 44% of term births (37 weeks) [5].

Many children with CP experience rigidity or weakness in muscles that tends inevitably towards fixed or irreversible deformities. The impairment of muscle activation and motor control has a negative impact on the performance of daily activities such as dressing, feeding, and playing, as well as on cognition [6]. Currently, CP stands as the leading cause of childhood disability. Therefore, the motor and cognitive rehabilitation of these patients is deemed essential from a perspective of necessity, employing a multidisciplinary approach. The goal is to stimulate the learning of functions that the patient would not be able to acquire on their own. Management and treatment strategies include orthotic systems, prosthetics, and aids of all kinds; neurosurgical interventions such as deep brain stimulation, intrathecal baclofen, and selective dorsal rhizotomy; botulinum toxin; pharmacological approaches; orthopedic surgery; and various rehabilitation interventions, including the use of aquatic environments as in hydro-kinesitherapy [7]. Neuromotor rehabilitation is the primary treatment aiming to stimulate the learning of motor functions the patient would not be able to acquire on their own. 

In recent years, the development of new technologies has facilitated the spread of rehabilitation through virtual reality (VR), involving technological tools that seem to bring about functional improvements [8]. These systems aim to assist the patient in rehabilitation procedures, ensuring that the actions performed, despite being repetitive and intensive, are accurate and personalized based on collected individual data [9]. They include exercises that impact the patient’s nervous system to interactively restore functions. This is achieved through real-time processing of motor and physiological data, always in accordance with training protocols based on neural principles, such as plasticity [9]. During neuromotor training using VR, which allows for prolonged attention retention due to highly motivating game stimuli for young patients, several factors come into play. These include goal-oriented tasks, repetition, and feedback, all of which stimulate neuronal plasticity, inducing better recovery [10]. Moreover, early intervention is crucial for recovery associated with neuroplasticity. Another interesting rehabilitative aspect is the ability to manipulate virtual environments to adapt them to the therapeutic needs and functional characteristics of the patient, both cognitive and motor. 

Previous studies have shown that rehabilitation treatments using VR can effectively stimulate and train lost motor function by inducing cortical neuronal reorganization and promoting neuronal plasticity. This may lead to positive outcomes, particularly in upper limb motor rehabilitation, exceeding those obtained with conventional rehabilitation alone [11,12,13]. Beyond CP, VR has demonstrated its usefulness in various rehabilitation settings. In post-stroke rehabilitation, VR systems offer immersive environments for motor relearning and functional recovery, allowing patients to engage in activities such as reaching, grasping, and walking within tailored virtual environments [14,15,16]. Similarly, traumatic brain injury rehabilitation uses VR technology to facilitate cognitive rehabilitation, including training attention, memory, and executive functions [17,18]. By simulating real-world scenarios and activities, VR allows people to improve cognitive and motor abilities in a safe and controlled environment. Furthermore, VR holds promise in spinal cord injury rehabilitation, providing opportunities for motor and sensory retraining, as well as psychological support through immersive experiences [19]. These diverse applications highlight the potential of VR to transform neuromotor rehabilitation, offering innovative and personalized interventions tailored to individuals’ unique neurological needs.

However, the primary objective of this review is to investigate the role of immersive VR in patients with CP, and the potential effects on both motor and cognitive outcomes.

## 2. Materials and Methods

We conducted this systematic review to understand the state of the art regarding the use of immersive VR in the rehabilitation of CP. This systematic review adhered to the Preferred Reporting Items for Systematic Reviews and Meta-Analyses (PRISMA) guidelines [20] and has been registered with the DOI (10.17605/OSF.IO/Y2DBS) in the Open Science Framework (OSF).

### 2.1. PICO Model

We employed the PICO (Population, Intervention, Comparison, and Outcome) model to shape our research question [21]. Our target population focuses on children affected by cerebral palsy. The investigated intervention was the state-of-the-art use of immersive VR in this patient group, encompassing all cognitive and motor domains. For the comparison, we focused on other interventions. The outcome was related to evidence from the use of immersive VR in CP.

### 2.2. Search Strategy and Eligibility Criteria

A review was conducted for all peer-reviewed articles published, using the following terms: ((“Virtual reality” OR “Immersive virtual reality” OR “Semi-immersive virtual reality”) AND (“Cerebral palsy” OR “Motor impairment” OR “Motor rehabilitation” OR “Cognitive impairment” OR “Cognitive rehabilitation”)). A review of currently published studies was performed in the following databases: PubMed, Embase, Cochrane Database of Systematic Reviews, RehabData, and Web of Science. All search results were screened by two blinded reviewers (MGM and MCV) to minimize the risk of bias (e.g., publication bias, delay bias, language bias). After screening based on titles and abstracts, the blind was opened and in case of disagreement, the other two reviewers were included in the decision process (GM, RSC). The list of articles was then refined for relevance, revised, and summarized, with the key topics identified from the summary based on the inclusion/exclusion criteria. 

The inclusion criteria were as follows: (i) focusing on children with CP; (ii) studies that described or investigated the use of immersive VR in CP; (iii) published in the English language; and (iv) published in a peer-reviewed journal. We excluded articles describing theoretical models, methodological approaches, algorithms, and basic technical descriptions. Additionally, we excluded (i) animal studies and (ii) and conference proceedings or reviews.

### 2.3. Assess Quality of Included Studies—Risk of Bias 

Each study was evaluated for methodological quality using the Newcastle–Ottawa Scale (NOS) [22], following the criteria outlined by the Cochrane Non-Randomized Studies Methods Working Group (Table 1).

Originally designed for observational studies, the NOS was adapted to assess the methodological quality of non-randomized interventional studies. Evaluation included key areas such as participant selection, comparability of groups, and assessment of outcomes. The NOS facilitated a systematic assessment of potential bias, providing insights into the strengths and limitations of the reviewed studies. Quality assessment was independently conducted by two authors (MGM and MCV), with the level of agreement between them calculated by a third author (GM). The detailed results of this evaluation are presented below.

The methodological quality of the included studies varied, with total scores ranging from 4 to 7. While some studies achieved relatively high scores [23,24,25,26,27,31,32,33,34,35,36], indicating good methodological rigor, others scored lower [28,29,30,37], suggesting potential limitations in study design or execution. Overall, the quality of evidence across the reviewed studies appears to be moderate to good, with most studies demonstrating satisfactory methodological quality according to the NOS criteria [22].

## 3. Results

After a preliminary search, 788 articles were found. After removing duplicates, 695 articles remained. Furthermore, 680 articles were excluded because they did not meet the inclusion criteria; thus, 15 articles remained (Figure 1, Table 2). Moreover, the agreement between the two reviewers (MGM and MV) was assessed using the kappa statistic. The kappa score, with an accepted threshold for substantial agreement set at >0.71, was interpreted to reflect excellent concordance between the reviewers. This criterion ensures a robust evaluation of the inter-rater reliability, emphasizing the achievement of a substantial level of agreement in the data extraction process. 

## 4. Key Finding

Of the fifteen included studies, many of them deal with the GRAIL system [23,24,25,26,27]; other studies used immersive VR systems like CAREN [28,29], Oculus Rift [30,31], and VR with a horse riding simulation [32,33]. Finally, two studies used VR with double tapis-roulant [34,35]. 

Bortone et al. demonstrated in an RCT that VR-assisted therapy is no less effective than conventional therapy, suggesting that it can be used as an alternative or complementary to conventional therapy [30]. The authors showed that the combination of treatment with immersive virtual environments and wearable haptic devices (VERA) using the HMD (Oculus Rift VK2) and conventional therapy in 28 children improved the kinesiological indices of linear path tracking and reach-and-grasp tasks. 

The feasibility, effectiveness, and significant non-inferiority of VR treatment compared to non-VR treatment were confirmed by Saussez et al. [36] in an RCT where 40 children with hemiparetic CP (20 in the experimental group, undergoing HABIT-ILE therapy in association with semi-immersive virtual reality (REAtouch), and 20 in the control group, undergoing therapy only with HABIT-ILE, showed significant improvements in gross motor function and ambulation. The REAtouch intervention showed significant improvements in the less affected hand, while the HABIT-ILE intervention showed significant improvements in the more affected hand. The use of a VR environment can create a motivational setting capable of providing feedback to subjects, training them, and/or challenging them outside their comfort zone, as demonstrated by Sloot et al. in their study of 20 children, where potential benefits of self-regulated walking and VR were observed, resulting in increased gait variability and speed [34]. The use of virtual reality allowed for significant improvements in motor functions, limb-specific abilities, and cognitive functions.

For motor functions, the improvement in gross motor skills was demonstrated by Gagliardi et al. following immersive VR treatment with The Gait Real-Time Analysis Interactive Lab (GRAIL system) in 16 children with CP, notably in standing, walking, running, and jumping abilities [23]. Besides gross motor skills, resilience also improved significantly, and progress was seen in hip and ankle functions. Consistent with these findings, Booth et al., through immersive VR treatment with a double-belt instrumented treadmill, showed improvements in ankle power, knee extension, and stride length in 22 children [35]. Moreover, the efficacy of the GRAIL system, an immersive VR tool, was also demonstrated by van Gelder et al., who showed an improvement in hip and/or knee extension [26]. The results of van der Krogt et al. also demonstrated the effectiveness and feasibility of immersive VR executed with the GRAIL system on independent treadmill walking in 20 children, although the walking speed was lower than natural walking speed [27].

Another significant contribution was made by Jung et al. [32], who observed improvements in gross motor function, mobility, and control through immersive VR treatment with a horse riding simulator, enhancing standing, running, and jumping abilities in 16 children with CP. Other authors, such as Chang et al., used the same tool, demonstrating that this training could promote mobility and balance control in 17 children, with positive effects on body composition (increase in height, lean mass, and skeletal muscle mass) [33].

Another immersive VR treatment, through the CAREN+ system, produced statistically significant results in a study by Ma et al. [29], where uphill walking showed reduced walking speed and step length, and increased pelvic tilt peak, ankle dorsiflexion, and hip flexion in 20 children with CP. Compared to typically developing children, those with CP exhibited decreased walking speed and step length, decreased peak hip abduction moment, increased static phase percentage, increased ankle dorsiflexion peak and knee flexion, and increased peak hip extension moment. Unlike previous studies, the virtual rehabilitation system CAREN+ was also used in the study by Barton et al. [28], demonstrating no significant inferiority of VR treatment compared to non-VR treatment, as demonstrated by Saussez et al. [36].

Regarding limb improvements through virtual reality, Shum et al. [31] demonstrated that immersive VR treatment through the Oculus Rift system in 17 children allowed improvements in bimanual movement symmetry in both groups using increased error feedback during virtual reality use. Bortone et al. [30], through treatment dedicated to upper limb manipulation rehabilitation with two wearable tactile interfaces for cutaneous feedback, two dedicated immersive serious games, and a graphical user interface, demonstrated the usability of the immersive VR tool in 20 children, divided into three clusters with three different levels of motor skills. All children had the opportunity to complete the experimental rehabilitation session, demonstrating that the VR system can be compatible with different levels of motor skills. Moreover, rehabilitation through virtual reality tools has also led to significant results in the cognitive domain, as demonstrated by the study of Nossa et al., who highlighted improvements in visuospatial abilities in children with CP through VR training performed using the GRAIL system [24]. The tool was statistically significant for the Labyrinth subtest (WISH-III). This subtest primarily measures planning ability, perceptual organization, visuomotor coordination, self-control, as well as motor performance and efficiency. These results are supported by the study of Biffi et al., where using the GRAIL system in 28 children resulted in improved performance in navigating a virtual maze [25].

## 5. Discussion

The aim of the present review was to investigate the role of immersive VR in patients with CP in terms of both cognitive and motor outcomes. Our studies highlighted that VR could be a useful tool to enhance functional outcomes in children suffering from PC. Rehabilitation based on VR, within a multimodal perspective of rehabilitation (motor, cognitive, and sensory), improves motor learning processes through implicit learning, concrete tasks, and focused attention [23,24,25,26,27]. Furthermore, directing the patient’s attention to the outcomes of their movements using sounds, visual stimuli, or point-earning, all possible in VR games, is more effective than focusing on the movement itself. The increased motivation not only allows for more repetitions but also intensifies the bioelectrical signals in the brain involved in neuroplasticity [38].

One of the strengths of this review is that it has synthesized the evidence on immersive VR in children with PCI, excluding non-immersive VR and serious games. It is important to consider that over the last few years the quality of the research on VR has greatly improved. In fact, at the beginning there was confusion between immersive and non-immersive VR, and with serious games/exergaming [39]. Today, however, it is known that immersive VR has very different technology requirements and principles of human interaction than non-immersive VR thanks to the sense of presence [40]. The continuous advancements in VR technology, coupled with concurrent reductions in associated costs, have facilitated the development of more user-friendly, beneficial, and accessible VR systems. A fundamental characteristic present in all VR systems is interaction. Interaction occurs through virtual environments created to allow users to interact with virtual objects within that environment. A wide range of visual interfaces is used to create varying degrees of immersion in a virtual environment, ranging from conventional monitors to head-mounted displays [39]. Immersion, where the user is surrounded by the virtual environment, is a crucial feature that distinguishes VR systems used in rehabilitation into immersive and non-immersive categories [41].

A completely different possibility is the serious game, very useful in providing feedback but different from VR as an adjunctive therapeutic tool [42]. In line with our review, recent manuscripts provide syntheses of VR’s efficacy focused specifically on immersive VR, for example, a recent paper by Demeco and colleagues [43].

The studies showed that the benefits of VR in rehabilitation are linked to the simultaneous stimulation of cognitive and motor processes. VR provides a rewarding and stimulating virtual environment for engaging in new experiences in recreational and enjoyable contexts. Additionally, VR encourages and motivates individuals to solve problems in a variety of situations [44]. Another fundamental characteristic of VR is to create a sense of actual presence in a simulated environment that is controllable by the user. Patients suitable for VR therapy are those who do not have visual or auditory impairments, do not have impaired cognitive function, and exhibit adequate cooperation and motivation. Furthermore, it is crucial that there is an absence of any significant joint contractures that would hinder exercises, and a lack of severe spasticity [45].

Our review demonstrates how immersive VR can be used in children with CP for a wide range of outcomes: motor (large motor, posture and walking, dexterity, symmetry of bimanual movements) and cognitive (visuo-spatial skills, planning skills, perceptual organization, visuo-motor coordination and self-control, rather than visuospatial memory), as well as non-cognitive/non-motor outcomes such as composition, such as increase in height, lean mass, and skeletal muscle mass [23,24,25,26,27,28,29,30,31,32,33,34,35,36].

With reference to an evaluation of the effectiveness of immersive VR, we can now say that the results are encouraging and certainly not inferior to conventional therapy. Very interesting are the degree of safety of this therapy and the easy integration with rehabilitation protocols, as demonstrated by the numerous objectives for which immersive VR is used. Indeed, our findings highlight that VR rehabilitation presents a promising approach to improving motor functionality in patients with CP, with potential differentiated impacts based on varying severity levels of the condition. By utilizing the Gross Motor Function Classification System (GMFCS) to identify CP severity levels and the Gross Motor Function Measure (GMFM) to assess motor function scores, some studies indicate that VR can positively influence motor rehabilitation and enhance patient outcomes concerning these parameters [23,32,33]. This approach enables a better evaluation of VR’s effectiveness across diverse clinical contexts and allows for therapy adaptation based on patients’ specific needs, thus contributing to optimizing rehabilitation outcomes across all severity levels of CP. It is important that randomized and controlled research studies are planned in the future with appropriate methodology and with a clear idea of the VR to be investigated for a specific objective.

Notably, immersive VR, unlike non-immersive reality, can also have major adverse events by generating significant human–scenario interactions. Cybersickness, or motion sickness, is a side effect of virtual reality experiences with the use of a headset that allows an immersive experience [45]. It results from a misalignment between sensory inputs. Cybersickness may include nausea, vomiting, headache, pallor, eye fatigue, disorientation, ataxia, and dizziness. It is a significant drawback for rehabilitation through VR systems and diminishes the treatment’s advantage. The effects of cybersickness can persist for several minutes and, depending on the severity, even for hours or days [46]. This diminishes public interest in VR, and symptoms may appear sometime after the VR experience. The most direct consequence is a decrease in the use of VR systems by users, simply because people tend to avoid what makes them feel unwell. Experiencing discomfort during a virtual reality session can also lead to distraction, resulting in a loss of presence and immersion. Ultimately, it may lead users to avoid behaviors causing cybersickness, effectively excluding part of the experience [47]. However, our review shows that out of 15 included studies, there were very few dropouts and unadministered sessions, which is in line with other studies with other devices. This shows that even though cybersickness is documented, it does not represent a real obstacle to the real usability, feasibility, and security of VR.

A further point to consider is that the medical devices to be used for clinical purposes should have a CE marking and not be VR or gaming devices developed for fun that are not suitable for a medical rehabilitation environment (e.g., Xbox, Nintendo Wii, PlayStation), even though these devices have been evaluated for research purposes [40].

Moreover, the implementation of VR in rehabilitation presents an interesting combination of costs and benefits. On the one hand, the implementation of this technology can involve significant initial costs, including those related to the purchase of specialized equipment and staff training [48]. However, it is important to consider that investments in VR may be offset by the long-term benefits offered by the technology itself. VR provides a highly customizable and immersive therapeutic environment, which can improve patient motivation and treatment adherence [49]. Furthermore, monitoring and recording data in real time allows therapists to customize and optimize the rehabilitation program based on the patient’s specific needs [17,18]. This can lead to more effective outcomes and faster recovery times, potentially reducing the long-term costs associated with managing chronic health conditions [48,49]. Therefore, although VR may represent a significant initial investment, its potential benefits in terms of improving the effectiveness and efficiency of rehabilitation may justify its use.

While our review highlights the potential benefits of immersive VR for children with CP, we acknowledge certain limitations in the studies analyzed. First, there is still ambiguity regarding the distinction between immersive and non-immersive VR, as well as serious games. Although our focus was on immersive VR, technological advancements have helped clarify these distinctions. Second, while VR shows promise in improving various outcomes, such as motor and cognitive skills, cybersickness presents a challenge that requires further investigation to understand its impact. Third, adherence to regulatory standards for VR devices in clinical settings is crucial to ensure safety and efficacy. Further research is needed to confirm the effectiveness of immersive VR in children with CP. Neurorehabilitation studies incorporating VR should provide clearer specifications regarding the type of device used, categorize it accurately, delineate the virtual scenario, and articulate the rehabilitative purpose for which it is employed. Lastly, the potential impact of immersive VR therapy on the vision of children with CP remains unaddressed in the literature, warranting further investigation.

## 6. Conclusions

This review underscores the potential of VR as a valuable tool for enhancing both motor and cognitive function in children with CP. The immersive and engaging virtual environments, coupled with audio-visual feedback, may enhance motor learning and functional recovery beyond what conventional therapy offers. However, the current ambiguity surrounding the classification of VR types (non-immersive, semi-immersive, or immersive) and the diversity of devices used in various studies necessitate further research with a more standardized methodology. Clarifying these factors will be essential to determine the optimal use of VR in improving outcomes for patients with CP.

## Figures and Tables

**Figure 1 brainsci-14-00490-f001:**
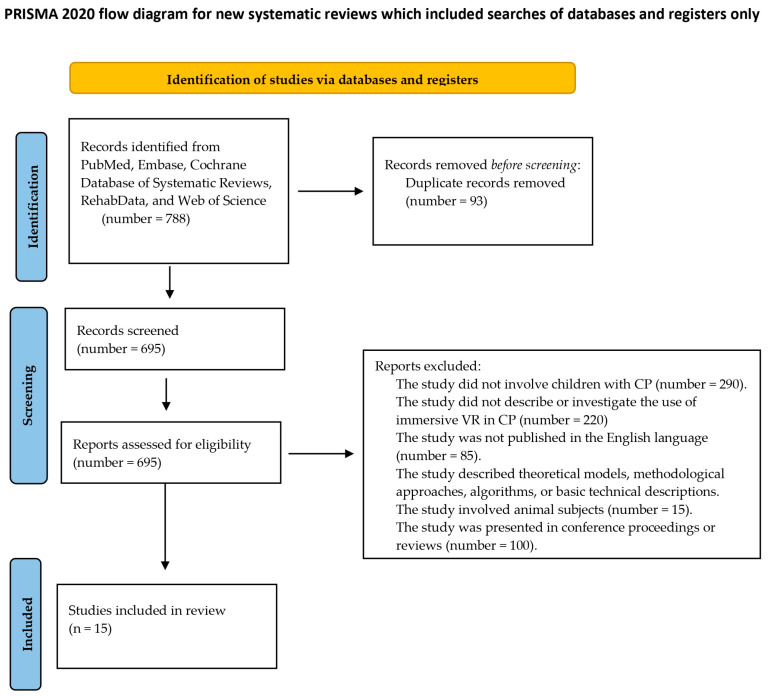
Flowchart of selected studies.

**Table 1 brainsci-14-00490-t001:** The NOS results.

Study	Selection	Comparability	Outcome Assessment	Total Score
Gagliardi et al. [23]	3	1	2	6
Nossa et al. [24]	3	2	2	7
Biffi et al. [25]	3	1	2	6
van Gelder et al. [26]	3	1	2	6
van der Krogt et al. [27]	3	1	2	6
Barton et al. [28]	1	2	1	4
Ma et al. [29]	2	1	2	5
Bortone et al. [30]	1	2	1	4
Shum et al. [31]	3	1	2	6
Jung et al. [32]	3	2	2	7
Chang et al. [33]	3	2	2	7
Sloot et al. [34]	3	2	2	7
Booth et al. [35]	3	1	2	6
Saussez et al. [36]	3	2	2	7
Bortone et al. [37]	2	1	2	5

**Table 2 brainsci-14-00490-t002:** Main studies about VR treatment in CP.

Ref.	Type of Studies	Participant	Intervention	Frequency and Duration	Primary Outcome	Secondary Outcome	Drop Out	Major Findings
Gagliardi et al. [23]	Pilot Study	16 children with bilateral CP diplegia SMI level I, II, and III (7–16 years); 10 males and 6 females	IVR using GRAIL system for exercises targeting walking and balance	One daily session lasting 30 min, 5 days a week (18 sessions)	GMFM 88 6MWT FAQ	The Sensewear Armband wearable device was used to measure energy expenditure	No drop-out	Motor skills, including standing, walking, running, and jumping, significantly improved, along with enhanced walking performance indicated by kinematic and kinetic parameters. Progress was observed in hip and ankle functions.
Nossa et al. [24]	Pilot Study	41 children (7–15 years): -14 TD 35 preterm spastic diplegia CP and SMI level I, II, and III.	GRAIL system Regular IVR training The IVR navigation Training	GC and GI underwent 18 daily sessions, each lasting 45 min	Corsi Block Test Subtest Labyrinth of WISC-III “Star-Maze” app	GMFCS MACS	No drop-out	All children with CP showed improved visuospatial abilities after both training courses, indicating the effectiveness of the VR programs. Overall, children with CP demonstrated enhanced performance and motor efficiency.
Biffi et al. [25]	Pilot Study	28 children: 15 with bilateral CP and SMI level I–III, (6–14 years: 11 males and 4 females) 13 TD (5 males and 8 females)	IVR GRAIL system for gait and balance	21 explorations of the maze, with 16 attempts to freely explore the environment plus five interposed trials	Corsi Block Test Subtest Labyrinth of WISC-III “Star-Maze” app on the GRAIL system	Raven’s progressive matrices	No drop-out	Both groups improved navigation skills in the virtual maze over trials. Typically developing participants quickly mastered maze navigation. Participants with CP navigated similarly once performance stabilized, suggesting minimal impact of motor impairment.
van Gelder et al. [26]	Clinical Study	27 children: 16 spastic CP and SMI level I–III (6–16 years) 11 TD (6–16 years)	IVR GRAIL system And 3D motion capture (Vicon, Oxford, UK)	Self-selected walking speed was assessed for the first 3 min without feedback, followed by feedback on knee extension and hip extension	HBM outputted 3D kinematic data, GPS, and MAP incorporating trunk kinematics		No drop-out	All the children, except one, improved hip and/or knee extension.
van der Krogt et al. [27]	Clinical Study	20 children: 9 with spastic CP, SMI level I or II (8–14 years) 11 TD (7 males and 4 females between 8 and 15 years old)	3 treatment: conventional gait lab; GRAIL system; indoor courtyard	Four different 3 min test trials were collected in random order	Various parameters including joint angles, gait velocity, step width, stance motion	GPS MAP The similarity to walking in the street, whether they could walk alone, preferred speed, and fatigue in walking	No drop-out	After training, all children walked independently on the treadmill. Step width and knee/ankle movements varied systematically in PCI; potentially clinically relevant. Walking speed in both labs was slower than natural.
Barton et al. [28]	Case Study	1 child with spastic CP diplegia, SMI level I (10 years old)	IRV Goblin Post Office game CAREN Vicon system. MATLAB’s CONVHULL The exercise takes place on the knees	Treatment of 30 min, 2 times a week, for 6 weeks (13 sessions)	Segmental Assessment of Trunk Control Gait Deviation Index	N/A	No	Both groups exhibited significant improvements, particularly in the least affected hand with the REAtouch intervention and in the most affected hand with the HABIT-ILE group. Additionally, there was no significant disadvantage in virtual reality (RV) treatment
Ma et al. [29]	Clinical Study	20 children: 10 spastic CP and SMI level I–II (6 males and 4 females: age 6–12 years) 10 children TD	IVR CAREN system 3D motion capture system	1 session	Joint kinematics, walking speed, peak pelvic tilt, ankle dorsiflexion, trunk rotation, stance phase, and ankle angle	Position of the center of pressure The position of the center of mass	No drop-out	During uphill walking, both groups slowed down and shortened steps, with increased pelvic tilt, ankle dorsiflexion, and hip flexion. Children with CP further reduced walking speed and step length, showing altered hip and ankle mechanics compared to TD children.
Bortone et al. [30]	Pilot Study Cross-over	8 children with neuromotor impairments: 3 CP SMI levels I–IV and MACS levels I–III 5 DD	IVR and wearable haptic devices (VERA) using the HMD (Oculus Rift VK2) in the first period + conventional therapy in the second	8 h (2 sessions per week for 4 weeks) of VERA rehabilitation before receiving conventional therapy	9-HPT	Zoia’s Protocol for DD Melbourne Assessment of unilateral upper limb function kinesiological assessment	No drop-out	Both conventional and VR-assisted therapies exhibited similar efficacy in improving kinesiological indices for specific tasks, suggesting VR therapy’s potential as a safe alternative or complement to conventional methods.
Shum et al. [31]	crossover counterbalanced design	17 children: 12 TD (13–21 years) 5 hemiplegic CP SMI level I-III	Bimanual treatment of the upper limb using Oculus Rift system, l’Oculus touch controller	The study was a single-session experiment	Symmetry Root-Mean-Squared Error (RMSE) in cm	Rom peak velocity per reach time to peak velocity movement smoothness trunk compensation	No drop-out	There were improvements in the symmetry of bimanual movements in both groups that used increased error feedback during the use of virtual reality.
Jung et al. [32]	Clinical Study	17 children with spastic CP diplegia SMI level I–IV EG: 10 CP (7 males and 3 females) CG: 7 CP (4 males and 3 females)	EG: IVR HRS and conventional Ph CG: home-based aerobic exercise and conventional Ph	Twice a week for a total of 16 sessions for both groups	GMFM	BIA PBS TUG	No drop-out	The study demonstrated that high-resistance strength training with virtual reality yielded positive effects on motor function, balance, mobility, and body composition in children with spastic CP, notably increasing skeletal muscle mass, without significant adverse events.
Chang et al. [33]	Pilot Study	16 children with CP SMI levels I–IV (5–17 years); 6 females and 10 males	IVR HRS	30 min twice a week over a period of 8 weeks (total of 16 sessions)	PBS GMFM-88 GMFM-66	N/A	No drop-out	Statistically significant improvements in PBS, GMFM-66, and GMFM-88 scores, particularly in standing and walking, were observed without any reported adverse events.
Sloot et al. [34]	Clinical Study	20 children: 9 with CP SMI level I or II (5 females and 4 males, age 8–14 years) 11 TD (4 females and 7 males aged 8–15 years)	GRAIL system and Three markers of movement to the pelvis, thighs, shanks, and feet	Participants familiarized with treadmill walking before conducting four random trials: walking at preferred speed with and without VR	Ground reaction force motion data 3D kinematics and kinetics Joint and segment angles Walking speed, stride length, stride time, step width, and stance percentage	The gait pattern; the ankle peak power; and work for the hip, knee, and ankle	No drop-out	The study suggests self-regulated and treadmill-induced walking, with or without VR, are interchangeable for gait analysis, with potential benefits such as increased walking speed variability during self-paced walking and VR’s motivational aspect akin to surface walking, providing feedback or challenges.
Booth et al. [35]	Clinical Study	22 children with spastic CP, SMI level I–II (age between 5 and 16 years old)	Double-belt instrumented treadmill Camera system with 26 retro-reflective markersHuman body model	1 session	Ankle power generation during pushing, knee extension, stride length, and biofeedback on aspects of gait	Stride length, knee extension, and ankle power	3 children	There were significant increases in ankle power and notable improvements in knee extension and stride length, which are clinically significant.
Saussez et al. [36]	RCT	40 children with hemiparetic CP-MACS I–III and GMFCS I–II (age 5–18 years): EG = 20 children CG = 20 children	EG: therapy with HABIT-ILE and SEMI IVR (Reatouch) CG: therapy with HABIT-ILE	EG: 53 h of HABIT-ILE and 37 h of REAtouch over two weeks, while the CG underwent 90 h of HABIT-ILE over the same duration	AHA	JTHFT BBT 6MWT ACTIVLIM-CP PEDI COPM	2 children	Both groups demonstrated significant improvements in most measures, with REAtouch showing efficacy in the less affected hand and HABIT-ILE in the more affected hand, suggesting REAtouch’s non-inferiority during HABIT-ILE compared to conventional intervention in children with unilateral cerebral palsy.
Bortone et al. [37]	Clinical Study	20 children EG: 3 children with CP (MACS I-III and GMFCS I–II) and 5 children with DD (age 7–14 years) CG: 8 children with TD (age 8–16 years) and 4 adults (age 24)	Two wearable haptic interfaces for cutaneous feedback, two dedicated immersive serious games for upper limbs	Each of the 4 levels were performed 3 times, for a total of 12 repetitions	Zoia’s protocol 9-HPT	A kinematic evaluation	1 child	The findings indicate the system’s compatibility with diverse motor skill levels, enabling patients to complete the experimental rehabilitation session, with performance varying according to the expected motor skills of distinct groups.

## Data Availability

Not applicable.

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
