# Peer review of "The Role of Immersive Virtual Reality Interventions in Pediatric Cerebral Palsy: A Systematic Review across Motor and Cognitive Domains"

_brainsci, 2024, doi:10.3390/brainsci14050490_

Round 1

Reviewer 1 Report

Comments and Suggestions for Authors

Thank you for the opportunity to review this manuscript. It is my pleasure to be involved with the research presented in this article. This study investigates the safety and efficacy of immersive virtual reality for children with cerebral palsy.

After carefully reviewing the article, I have a few suggestions to make to the authors that could improve the overall quality of the work.

- I consider that it could benefit from a clearer structuring in the abstract. I believe this modification would help readers to better understand the main findings and conclusions of your study.

To achieve this, I suggest dividing the abstract into clearly defined sections, highlighting the most relevant results in each section. For example, you might consider including the following sections:background, methods, results and conclusions.

To recapitulate the main conclusions of the study, emphasising the importance of immersive virtual reality as a rehabilitation tool in conventional cerebral palsy programmes.

I strongly believe that this structuring would improve the clarity and accessibility of your article, allowing readers to easily understand the most important aspects of your research.

- Introduction: , it would be useful to add a short section explaining how virtual reality has been used in other rehabilitation contexts, beyond cerebral palsy, to further contextualise the use of this technology in the field of neuromotor rehabilitation.

As for the reference to the study by Meyer-Heim and van Hedel (2013), it is presented as a relevant previous study supporting the efficacy of VR in motor rehabilitation, but it would be useful to mention other previous studies that have also investigated the use of VR in the context of cerebral palsy. This would provide a broader overview of the existing literature on the topic.

Methods:

-in material and methods it is commented that the PICO criteria are followed but not specified. Then the inclusion criteria are specified...but it does not follow the PICO criteria. This is confusing.

-“After a preliminary search, 788 articles were found. After removing duplicates, 695 articles 113 remained. Furthermore, 680 articles were excluded because they did not meet the inclusion criteria, thus, 15 articles remained (figure 1, table 1)….

this text should go in results.In my view the authors are misleading in the sections results and material and methods. Please revise. It should also be specified how the methodological quality is to be assessed.

- Figure 1 should be carried out again. There are different letter formats, several points have been joined as not in English language and theoretical models......

also a loose "n"...please do it again

-figure 1 should be in the results section and referenced in the results section, not in the material and methods section. Table 1 should also not be mentioned in the methods part, but in the results part.

- the last column of table 1 should specify the effect size of the main outcomes. This table should be positioned in the results table.

- Results: the results section should start with the number of articles obtained and how many were left in the end.....

- The tables of the evaluated results would be missing in order to see the methodological quality of the articles: Cochrane Risk of Bias (RoB) tool, PEDro scale, GRADE (Grading of Recommendations Assessment, Development and Evaluation) …depending on the type of design.

- Discussion section: In the discussion section, please comment on the limitations encountered.

- Conclusions: “the primary objective of this review is to investigate the role of immersive VR  in patients with Cerebral Palsy, and the potential effects on both motor and cognitive outcomes” …I don't see that the conclusion specifies the main conclusions according to the specified objective: what were the main cognitive and motor effects?

- The authors have commented that they followed the PRISMA guidelines, and I suggest that they revise the document to follow them more closely.

Author Response

Thank you for the opportunity to review this manuscript. It is my pleasure to be involved with the research presented in this article. This study investigates the safety and efficacy of immersive virtual reality for children with cerebral palsy. 

After carefully reviewing the article, I have a few suggestions to make to the authors that could improve the overall quality of the work. 

Thank you for your feedback. We have implemented the suggested changes in the text to make it better.  

- I consider that it could benefit from a clearer structuring in the abstract. I believe this modification would help readers to better understand the main findings and conclusions of your study. To achieve this, I suggest dividing the abstract into clearly defined sections, highlighting the most relevant results in each section. For example, you might consider including the following sections:background, methods, results and conclusions. 

Thanks for your comment. We agree that restructuring the abstract into clearly defined sections can improve readability and understanding, so we have divided it into sections as suggested. 

To recapitulate the main conclusions of the study, emphasising the importance of immersive virtual reality as a rehabilitation tool in conventional cerebral palsy programmes. 

Thank you for your comment. We have revised the conclusions of the abstract in line with your suggestions. We have emphasized the significance of immersive virtual reality as a rehabilitation tool in conventional cerebral palsy programs. 

I strongly believe that this structuring would improve the clarity and accessibility of your article, allowing readers to easily understand the most important aspects of your research. 

Thanks for your insight. We appreciate your valuable input and have incorporated these suggestions to ensure that readers can easily grasp key aspects of our research. 

- Introduction: it would be useful to add a short section explaining how virtual reality has been used in other rehabilitation contexts, beyond cerebral palsy, to further contextualise the use of this technology in the field of neuromotor rehabilitation.  

We thank the reviewer for his/her valuable feedback, we have included in the introductory section an in-depth analysis on the use of virtual reality in other rehabilitation contexts, in addition to cerebral palsy. 

As for the reference to the study by Meyer-Heim and van Hedel (2013), it is presented as a relevant previous study supporting the efficacy of VR in motor rehabilitation, but it would be useful to mention other previous studies that have also investigated the use of VR in the context of cerebral palsy. This would provide a broader overview of the existing literature on the topic. 

Thanks for your insight, we have added more references as requested. 

Methods: 

-in material and methods it is commented that the PICO criteria are followed but not specified. Then the inclusion criteria are specified...but it does not follow the PICO criteria. This is confusing. 

Thank you to the reviewer for requesting clarity. We have revised the paragraph and acknowledged the importance of making the PICO model and the search strategy more apparent. Therefore, we have divided the paragraph into separate sub-sections to clarify how our study follows the PICO model and to provide detailed descriptions of our search strategy. 

-“After a preliminary search, 788 articles were found. After removing duplicates, 695 articles 113 remained. Furthermore, 680 articles were excluded because they did not meet the inclusion criteria, thus, 15 articles remained (figure 1, table 1)….this text should go in results. In my view the authors are misleading in the sections results and material and methods. Please revise.  

Thank you for your feedback. We appreciate your input and have addressed your concern by relocating the mentioned paragraph to the Results section, as suggested. 

It should also be specified how the methodological quality is to be assessed. 

Thank you for raising this issue. We have addressed it by adding a clarification regarding the assessment of methodological quality. Specifically, we included information about the Newcastle-Ottawa Scale (NOS) that we used to assess the quality of the studies. We noted that the quality ranged from moderate to good. 

- Figure 1 should be carried out again. There are different letter formats, several points have been joined as not in English language and theoretical models...... 

also a loose "n"...please do it again. 

Thank you for the comment. We revised the figure, as suggested.  

-figure 1 should be in the results section and referenced in the results section, not in the material and methods section. Table 1 should also not be mentioned in the methods part, but in the results part. - the last column of table 1 should specify the effect size of the main outcomes. This table should be positioned in the results table. 

Thank you for your comment and attention to detail. We have  included the effect size in Table 1 and carefully checked the availability of this measure in our data. Unfortunately, we are unable to provide the effect size for the main outcomes as it has not been calculated or is not available in the data reported in the articles included in the study. However, we appreciate the suggestion and are open to further comments or questions. Additionally, we have moved the figure and table to the Results section as suggested. 

- Results: the results section should start with the number of articles obtained and how many were left in the end..... 

Thank you for your comment. We have modified this part according to your suggestions. 

- The tables of the evaluated results would be missing in order to see the methodological quality of the articles: Cochrane Risk of Bias (RoB) tool, PEDro scale, GRADE (Grading of Recommendations Assessment, Development and Evaluation) …depending on the type of design. 

Thank you for your comment. To assess the quality of the studies, we utilized the Newcastle-Ottawa Scale (NOS), as many of the studies are pilot or clinical in nature and the NOS was deemed suitable for evaluating their methodological quality.  

- Discussion section: In the discussion section, please comment on the limitations encountered. 

We appreciate your suggestion, and as advised, we have incorporated the limitations into the final part of the discussion section. 

- Conclusions: “the primary objective of this review is to investigate the role of immersive VR  in patients with Cerebral Palsy, and the potential effects on both motor and cognitive outcomes” …I don't see that the conclusion specifies the main conclusions according to the specified objective: what were the main cognitive and motor effects? 

Thank you for your feedback. We have revised the conclusions to better align with the specified objective of investigating the role of immersive VR in patients with Cerebral Palsy and its potential effects on both motor and cognitive outcomes. 

- The authors have commented that they followed the PRISMA guidelines, and I suggest that they revise the document to follow them more closely. 

Thank you for your suggestion. We have revised the entire manuscript to better align with the PRISMA guidelines. This includes adding new sections such as the assessment of study quality and revising various sections for clarity and adherence to the guidelines. 

Reviewer 2 Report

Comments and Suggestions for Authors

This study explored the application of immersive virtual reality technology in the rehabilitation of children with cerebral palsy. The authors searched for relevant clinical studies that have been published. Although there was some heterogeneity among the studies, they still demonstrated the value of immersive virtual reality used in cerebral palsy. However, the research did not find immersive virtual reality technology to be superior to conventional rehabilitation. Considering that the devices generally require high economic investment, please add the economic-cost analysis of immersive virtual reality therapy. Additionally, considering the age of children with cerebral palsy, whether immersive virtual reality therapy would affect their vision was not addressed in the literature. This aspect should be addressed as well.

Author Response

This study explored the application of immersive virtual reality technology in the rehabilitation of children with cerebral palsy. The authors searched for relevant clinical studies that have been published. Although there was some heterogeneity among the studies, they still demonstrated the value of immersive virtual reality used in cerebral palsy. However, the research did not find immersive virtual reality technology to be superior to conventional rehabilitation.  

Thank you for your comment. We appreciate your acknowledgment of our study exploring the application of immersive virtual reality technology in the rehabilitation of children with cerebral palsy. 

Considering that the devices generally require high economic investment, please add the economic-cost analysis of immersive virtual reality therapy.  

Thank you for emphasizing this aspect. We agree, and we have added the economic-cost analysis of immersive virtual reality therapy in the discussion section. 

Additionally, considering the age of children with cerebral palsy, whether immersive virtual reality therapy would affect their vision was not addressed in the literature. This aspect should be addressed as well. 

Thank you for bringing up this important aspect. We have incorporated this issue into the limitations section of our discussion, highlighting the need for further investigation into its effects. 

Reviewer 3 Report

Comments and Suggestions for Authors

The manuscript is well organized. One correction is recommended. It would be beneficial to include a more detailed discussion of the features and challenges of VR rehabilitation in the introduction. Additionally, it may be beneficial to include some relevant literature. Finally, it would be advantageous to include a discussion of the extent to which VR rehabilitation can address specific GMFCS levels or GMFM scores for cerebral palsy.

Author Response

The manuscript is well organized. One correction is recommended.  

Thank you for the comments; we have incorporated the changes to enhance the quality of the study. 

It would be beneficial to include a more detailed discussion of the features and challenges of VR rehabilitation in the introduction. Additionally, it may be beneficial to include some relevant literature. 

Thank you for the suggestion. We have extended the discussion on VR rehabilitation in the introduction, including its features and challenges. Additionally, we have incorporated references as requested. 

Finally, it would be advantageous to include a discussion of the extent to which VR rehabilitation can address specific GMFCS levels or GMFM scores for cerebral palsy. 

Thank you for highlighting this aspect. We have included a discussion on the extent to which VR rehabilitation can address specific GMFCS levels or GMFM scores for cerebral palsy in the revised manuscript.